SOFTWARE

# Measuring, visualizing, and diagnosing reference bias with biastools

Mao-Jan Lin[1*], Sheila Iyer[1], Nae-Chyun Chen[1] and Ben Langmead[1*]

*Correspondence:
mlin77@jhu.edu;
langmea@cs.jhu.edu

[1] Department of Computer
Science, Johns Hopkins
University, Baltimore, USA

## Abstract

Many bioinformatics methods seek to reduce reference bias, but no methods exist to comprehensively measure it. `Biastools` analyzes and categorizes instances of reference bias. It works in various scenarios: when the donor's variants are known and reads are simulated; when donor variants are known and reads are real; and when variants are unknown and reads are real. Using `biastools`, we observe that more inclusive graph genomes result in fewer biased sites. We find that end-to-end alignment reduces bias at indels relative to local aligners. Finally, we use `biastools` to characterize how T2T references improve large-scale bias.

**Keywords:** Reference bias, Sequence alignment, Pangenomics

## Background

Most sequencing data analyses start by aligning sequencing reads to a reference genome. This strategy comes with a drawback called *reference bias*. The aligner tends to miss alignments or report incorrect alignments for reads containing non-reference alleles. This can lead to confounded measurements and incorrect results, especially for analyses of hypervariable regions [4], allele-specific effects [10, 33, 34, 38], ancient DNA analysis [17, 26], or epigenenomic signals [16].

Recent tools seek to reduce this bias by indexing collections of reference genome sequences, i.e., pangenomes. By including many known genetic variants in the pangenome, such methods remove alignment penalties incurred by known alternate alleles. This has spurred research in indexing graphs (e.g., the definition and use of Wheeler Graphs [13, 35]) and repetitive collections of strings, e.g., *r*-index [21] and hybrid indexes [37]. These ideas are used in practical tools like HISAT2 [20], VG [14] and VG-Giraffe [35]. Mitigating reference bias is also the stated motivation for the Human Pangenome Reference Consortium's project to create a human pangenome [25].

However, the topic of "reference bias" itself — what it means and how it happens — has received comparatively little attention. Studies proposing bias-reducing tools have

evaluated and visualized reference bias in divergent ways. There are no standard tools or metrics, and no methods exist to trace specific causes of reference bias events.

We present `Biastools`, a tool for measuring and diagnosing reference bias in datasets from diploid individuals such as humans. In its `simulate` mode, `biastools` enables users to set up and run simulation experiments to (a) compare different alignment programs and reference representations in terms of the bias they yield, and (b) categorize instances of reference bias according to their cause, which might be primarily due to genetic differences, repetitiveness, local coordinate ambiguity due to gaps, or other causes. In its `predict` mode, `biastools` enables users to analyze real sequencing datasets derived from donors with known genetic variants, both quantifying the overall level of reference bias and predicting which specific sites are most affected by bias. In its `scan` mode, `biastools` enables users to analyze real sequencing datasets from individuals with no foreknowledge of their genetic variants, identifying regions of higher reference bias.

We use `biastools` to study reference bias in various scenarios, including using aligners like Bowtie 2 [22], BWA-MEM [24] and the pangenome graph aligner VG Giraffe [35]. Our results support previous studies that found that including more variants in a pangenome graph reference reduces reference bias [6, 30]. Interestingly, we also find that end-to-end alignment modes of popular tools like Bowtie 2 and BWA-MEM (a local aligner by default, but with the ability to penalize non-end-to-end alignments) are particularly effective in reducing bias at insertions and deletions. By contrast, aligners that favor local alignments, with no penalty on "soft clipping," exhibit more bias around gaps. Finally, we found that applying `biastools`'s `scan` mode revealed large-scale differences in reference bias observed using only the GRCh38 assembly [7] versus when using the combined benefits of both the GRCh38 and the T2T-CHM13 [27] assemblies.

## Results

Ideally, a read aligner would map each read to its true point of origin with no bias toward one haplotype or the other. Also, an ideal method for analyzing read alignments and tallying the reference (REF) and alternate (ALT) alleles covering a given site would do so without introducing bias. However, real aligners, reference genomes and assignment methods are imperfect, and several factors interact to produce distinct reference-bias signatures. We describe how `biastools` can measure and plot reference bias. We focus on bias in the context of diploid individuals (i.e., human) being sequenced using high-quality short reads, e.g., from Illumina instruments.

### Measuring sources of bias in simulation

We performed a simulation experiment using `biastools`'s `simulate` mode, detailed in the "Methods" "Biastools workflow" section. We started from a Variant Call Format (VCF) file describing HG002's variants as determined by the Q100 project [31, 32], a collaboration between the Telomere-to-Telomere (T2T) consortium, Human Pangenome Reference Consortium (HPRC), and Genome in a Bottle (GIAB) project. We generated a diploid personalized reference genome for HG002 using `bcftools consensus`. We used `biastools --simulate`, which in turn uses `mason2` [19], to simulate Illumina-like whole genome sequencing (WGS) data to a total of ~30× average coverage,

taking ∼15× evenly from the two haplotypes. We used standard read aligners including Bowtie 2 [22], BWA-MEM [24], and Minimap 2 [23] to align to the GRCh38 reference genome [7]. We used VG Giraffe [35] to align to various graph pangenomes.

### Types of allelic balance

After simulation and alignment, we measured three types of allelic balance at each heterozygous (HET) variant site (Fig. 1). We measured ***simulation balance*** (SB) as the proportion of simulated reads overlapping the HET that originated from the REF-carrying haplotype. SB is computed purely from the simulator output; the simulator annotates reads with their haplotype and point of origin. We measured ***mapping balance*** (MB) as the allelic balance at each HET site considering only the reads that both truly originated from the HET (as reported by the simulator) and that overlapped it after read alignment. An overlapping read that originated from the REF-carrying haplotype contributes a REF allele, and likewise for an ALT-carrying read and ALT allele. MB ignores fine-grained details about how individual bases line up to the HET site in the pileup. Note that the *simulation balance* and *mapping balance* both use information from the simulator.

Finally, we measured ***assignment balance*** (AB) as the allelic balance after using an assignment algorithm to determine the haplotype of origin for each read overlapping the HET site. This does not make use of information from the simulator, and so can be measured for real reads as well as simulated ones. Assignment balance depends on the particular algorithm used to assign alignments to haplotypes. We tried two distinct algorithms, a "naive" assignment algorithm and a "context-aware" algorithm. The naive algorithm simply examines the nucleotides from each read that align across the HET site and computes a ratio according to how many of those sequences matched the REF allele

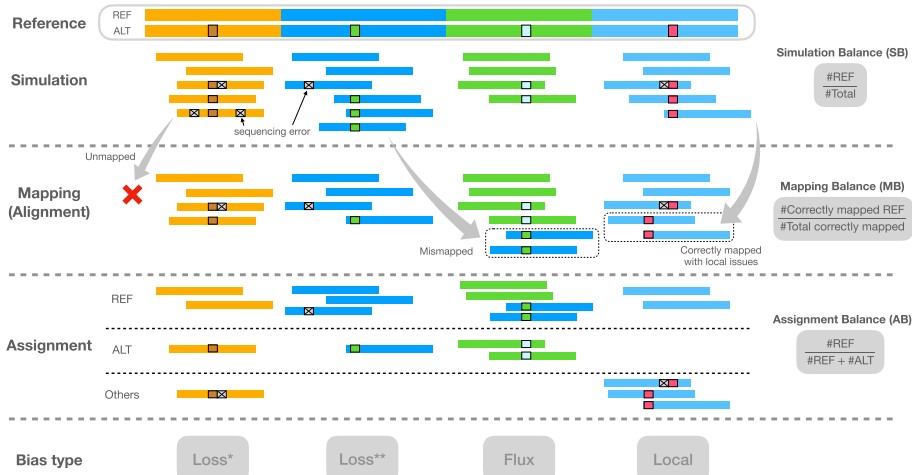

**Fig. 1** Illustration of the types of balance measurement — SB, MB, and AB — with respect to read simulation, read mapping, and halotype assignment. Note that the mismapped reads are excluded when calculating MB, and the reads assigned "Others" are also excluded when calculating AB. Columns indicate distinct types of bias event. "Loss*" indicates a bias event due to reads with ALT alleles failing to align. "Loss**" indicates a bias event due to reads mapping elsewhere than their true point of origin. "Flux" indicates bias from gaining mismapped reads from other sites. "Local" indicates that local repeat content, as well as sequencing errors, combine to make a gap placement ambiguous

versus how many matched the ALT allele. That is, the naive algorithm trusts that the aligner is correct and precise in how it places each base in the alignment and pileup.

The context-aware algorithm, on the other hand, does not trust the aligner's decisions, instead revisiting and possibly changing those decisions in light of all the alignments and the ploidy of the donor. It is a multi-part algorithm that decides whether each read is contributing a REF or ALT allele, or whether to exclude the read from consideration for lack of context. Assignment algorithms are detailed in the "Methods" "Assignment method" section.

### Types of reference bias

To categorize instances of reference biases, we computed these combinations of simulation balance (SB), mapping balance (MB) and assignment balance (AB):

- Normalized mapping balance (NMB) ≡ MB - SB. NMB > 0 implies that mapping creates more bias toward the REF allele compared to simulation, while NMB < 0 means mapping creates bias toward ALT.
- Normalized assignment balance (NAB) ≡ AB - SB. NAB > 0 implies that alignment and assignment together create more bias toward the REF allele compared to simulation, while NAB < 0 means mapping and assignment create bias toward ALT.

To demonstrate the utility of these measures, we examined the read alignments produced by Bowtie 2. We measured and plotted allelic balance at HET sites according to their NMB (horizontal) and NAB (vertical) (Fig. 2). Since SNVs and gaps exhibited distinct bias profiles, we plotted them separately. In this plot, HET sites with little or no bias will appear close to the origin. We called sites "balanced" and colored them green if they were within ±0.1 of 0 for NMB and NAB.

We next categorized HET sites that appeared far from the origin and along the diagonal (colored orange), the bulk of which were in the upper-right quadrant. Proximity to the diagonal indicates MB and AB are equally distant from SB. We inferred that this bias signature was likely introduced in the mapping stage, when reads systematically failed to align to the ALT-carrying haplotype. We called this "loss" bias. Most loss events appear in the upper right (as opposed to the lower left) because the ALT allele is usually harder for the aligner to map across, causing the aligner to fail more often.

We next categorized HET sites that were vertically above or below the origin. These sites had near-zero NMB, meaning that mapping did not introduce significant bias. The combination of near-zero NBM with non-zero NAB indicates that the reads overlapping the site are roughly evenly drawn from the REF and ALT alleles, but that the assignment algorithm has a bias in which allele it assigns. For points above the origin, there is a bias toward the REF allele after assignment.

We further divided these into "flux" and "local" events. Flux events (colored blue) involve reads with low mapping quality, indicating that the read aligner had nearly-equally-good choices for where to map these reads. Such reads may be placed incorrectly, leading to the true evidence for REFs and ALTs being spread (and averaged) over many copies of a repeat. Flux events were more common for SNVs and rarer for insertions and deletions.

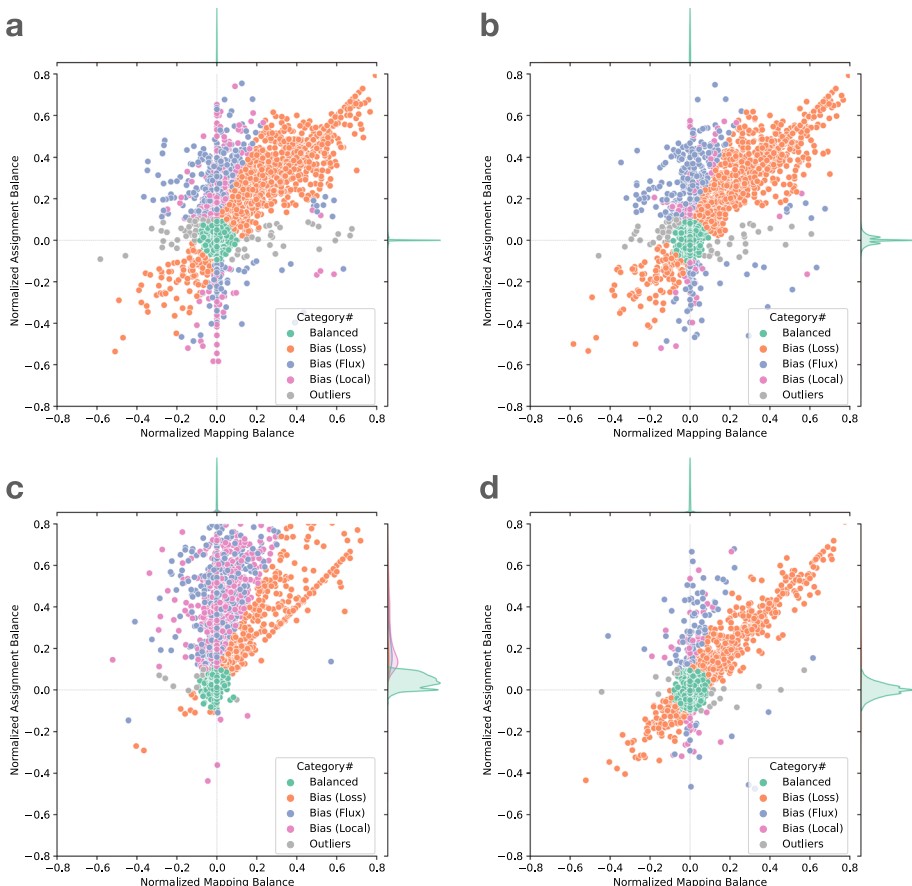

**Fig. 2** Normalized mapping balance to normalized assignment balance (NMB-NAB) plot of **a** SNV sites with naive assignment method, **b** SNV sites with context-aware assignment method, **c** insertion and deletion sites with naive assignment method, and **d** insertion and deletion sites with context-aware assignment method. Each dot represent a variant site in HG002 chromosome 20. The simulated reads are aligned using Bowtie 2 and default parameters. The balance and bias subcategories are classified based on the position of the dots ("Biased-site classification" section). For visual clarity, sites with no correctly-mapped REF reads are omitted; the full plot including these sites is available as Additional file 1: Fig. S1

Local events (colored purple) are those where the evidence comes from mostly high-mapping-quality reads. In these cases, we hypothesized that the bias was caused by the assignment step. When using the naive assignment method (Fig. 2a, c), most local bias events were caused by short tandem repeats, which created many equally good gap placements. Out of 3228 local bias events including SNVs and gaps, 2561 (79%) were at sites annotated by Repeatmasker. One thousand twelve of these sites were in Simple repeats (micro-satellites), 302 were in LINEs, and 934 were in SINEs.

When gap placement decisions are not consistent from read to read, this interferes with correct tallying of REF and ALT evidence and contributes to bias. This bias can potentially be avoided post facto by reconsidering and modifying the base-by-base alignments in light of the expected ploidy of the donor and the other alignments. This is the goal of past work on "local realignment" or "indel realignment," sometimes implemented in standalone tools [1, 18] or as components of larger variant-calling systems [8, 11].

A small number of sites did not belong to any of the above categories, and we called these "outliers" (colored gray). These can result from the co-occurrence of multiple of the above causes. A visual representation of all these categories is shown in the "Biased-site classification" section.

### Observations on local bias

Comparing Fig. 2 panels a and c (naive assignment) versus panels b and d (context-aware assignment), we observed that the context-aware method yielded fewer local-bias events compared to the naive method, especially for insertions and deletions. This was expected, since gap-placement ambiguity can cause the aligner to place a gap in a position that differs from its VCF position. The context-aware method avoids this by disregarding the aligner's gap decisions and scanning reads directly for variant sequences. Further, we stratified panels c and d by the length of the gaps (Additional file 1: Fig. S2). The three rows from top to bottom show the gaps longer than 10, 20, and 50 bases assigned by naive or context-aware method. It can be seen that the longer the gaps, the higher the ratio of variants are classified as "local" or "flux" bias in naive assignment. On the other hand, the context-aware method successfully classified the majority of the variants into "balanced" or "loss" in all scenarios.

We also observed that the context-aware method did not totally avoid local bias (Fig. 2b, d). Since this method requires that a substring of the read have an exact match to the REF or ALT allele at the site ("Assignment method" section), sequencing errors can affect the assignment balance either by artificially boosting the evidence for REF or ALT (if an error spurious creates a match), or more frequently by attenuating the evidence (if an error disrupts a match). This effect is more severe for longer insertions or deletions, since more opportunities exist for a position to mismatch. For long insertions, we expect the shorter REF allele to be less vulnerable to disruption by sequencing errors and so to be over-represented. For long deletions, we expect the ALT allele to be over-represented.

When multiple variants are situated near each other with respect to the reference, the read aligner can make decisions that cause context-aware assignment to fail. This can happen when a collection of nearby variants including gaps can be "explained" using fewer gaps and mismatches, causing portions of the read to shift with respect to the reference. An example is presented in Additional file 1: Fig. S3. The shifting is more likely to happen in the ALT allele, whereas sequencing errors happen roughly evenly in REF and ALT haplotypes.

### Visualizing bias for indels

We evaluated reference bias as a function of insertion and deletion length using the bias-by-allele-length plot (Fig. 3), modeled on a plot made in previous publications [9, 14, 35]. Here, the vertical axis is the ratio of alternate alleles observed spanning HET sites. That is, the vertical axis is the ratio ALT/(ALT+REF), where ALT and REF refer to the number of reads supporting the alternate and reference alleles respectively. For SNVs (length = 0), all measurements were well centered on 0.5. The naive assignment method (red) exhibited substantial bias across indel lengths, whereas both mapping balance (orange) and balance from context-aware assignment (green) stayed close to the simulation

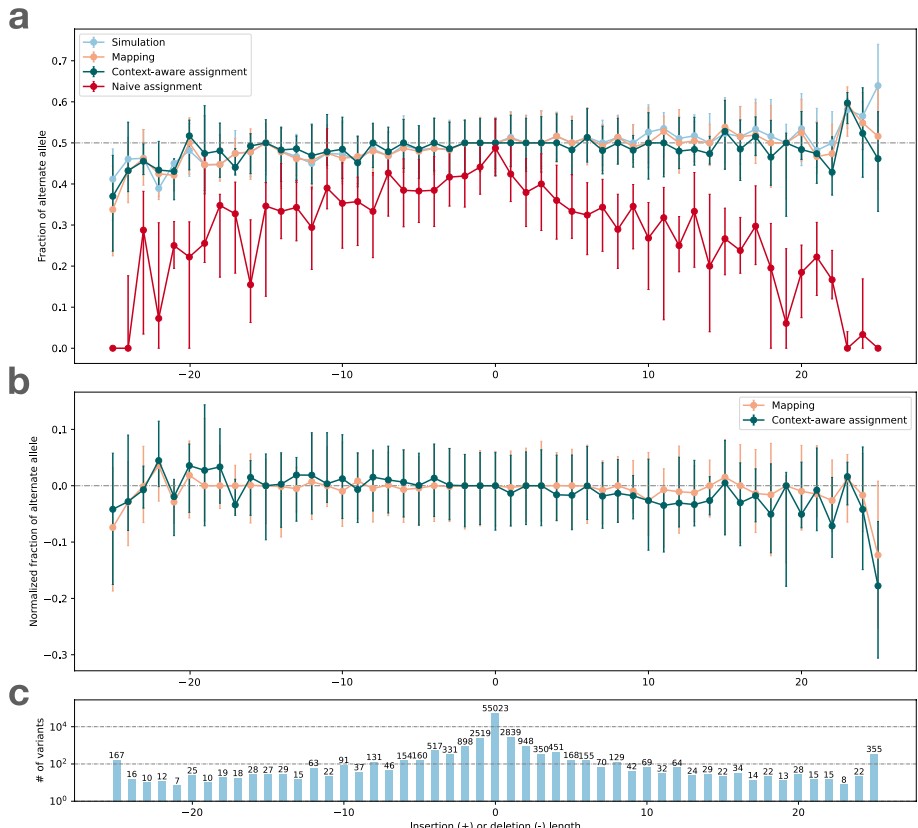

**Fig. 3** Bias-by-allele-length plots if we consider only Simulation Balance (blue), Mapping Balance (orange), Assignment Balance using context-aware assignment (green), and the same using naive assignment (red). Variant length varies along the *x*-axis, with positive values standing for insertion and negative values for deletions, and 0 for SNVs. The alignment is done by Bowtie 2 on HG002 simulated data. Top: Balance for all four measures. Dots represent median of the distribution and the whiskers indicate the first and third quartiles. Middle: Zoom-in on Mapping Balance and context-aware Assignment Bias with data normalized by subtracting median SB in each stratum. Bottom: number of variants with each length. Gaps exceeding 25 bp are collapsed into the −25 or 25 strata

balance. This occurs for the same reasons that we see more local bias events for the naive assignment method in Fig. 2.

## Measuring bias across aligners

We performed the above analysis using multiple read aligners, including Bowtie 2 (in its default end-to-end alignment mode) [22], BWA-MEM [24], BWA-MEM with option "`-L 30`" (to encourage end-to-end alignment) and the VG Giraffe graph aligner [35]. For VG Giraffe, we performed alignment using four different indexes.

- Giraffe-linear: a graph consisting only of the linear reference genome GRCh38 [7].
- Giraffe-major: a graph consisting of the GRCh38 reference but with major alleles added. With the addition of the major alleles, the graph contains 1,998,961 polymorphic sites.
- Giraffe-pop5: A graph consisting of all the variants from 5 pre-built haplotype genomes based on the "RandFlow-LD" pangenome used in the Reference Flow study

[6]. Each haplotype genome is based on a 1000 Genomes Project (1KGP) super-population. At each polymorphic site, the ALT allele is chosen with probability equal to its allele frequency. Linkage disequilibrium is preserved for each 1000 bp chunk. There are total 6,461,708 polymorphic sites across the 5 pre-built haplotype genomes combined.

- Giraffe-1KGP: A graph containing all the phase-3 variants from the 1KGP with allele frequency greater than 0.01, using GRCh38 as the reference. This graph contains a total of 13,511,768 polymorphic sites.

While Giraffe-linear uses the Giraffe graph aligner, the "graph" consists of a single linear genome in that case. The linear and major indexes serve as baselines to highlight how the inclusion of more variation (i.e., for Giraffe-pop5 and Giraffe-1KGP) impacts bias.

These experiments use the same simulated HG002 WGS dataset as in the previous section. In all cases, we used the context-aware assignment method to analyze allelic balance with respect to Q100 project-called variants for HG002 chromosome 20. Table 1 tallies and categorizes reference-bias events at chromosome-20 HET sites using the same classification strategy as in Fig. 2. The only category where aligners produced substantially different tallies was "loss," consistent with this category being directly related to the mapping of reads. Since Bowtie 2's default alignment mode is end-to-end alignment (which does not perform soft clipping) whereas the default mode for all other tools was local alignment (allowing soft clipping), we hypothesized that end-to-end alignment was a less biased strategy for gaps. To test this, we included results for BWA-MEM with the `-L 30` option, which increases the threshold for clipping from its default of `-L 5`. Specifically, BWA-MEM allows clipping only in cases where the increase in alignment score is greater than the number specified with `-L`. Consistent with our hypothesis, BWA-MEM with the `-L 30` option achieved the most balanced events for gaps compared to all other methods, including the end-to-end aligner, Bowtie 2, which achieved the second-most. The difference between the BWA-MEM modes

**Table 1** Number of balanced sites and different categories of biased sites on chromosome 20. The simulated WGS reads of HG002 are aligned by 8 different tools. The best results of balanced and Bias "Loss" are marked in bold and italic. The second best results are marked in bold

|   |   | Bowtie 2 | BWA-MEM | BWA-MEM (-L 30) | Minimap2 | Giraffe-linear | Giraffe-major | Giraffe-pop5 | Giraffe-1KGP |
|---|---|---|---|---|---|---|---|---|---|
| SNV | Balanced | 52,173 | 52,133 | 52,336 | 52,148 | 52,267 | 52,267 | **52,353** | *52,402* |
|  | Bias (loss) | 2108 | 2084 | 1881 | 2028 | 1924 | 1911 | **1812** | *1752* |
|  | Bias (flux) | *495* | 569 | 594 | 581 | 566 | 561 | **548** | 573 |
|  | Bias (local) | **169** | 177 | *158* | 186 | 183 | 210 | 228 | 211 |
|  | Outliers | 78 | **60** | *54* | 80 | 83 | 74 | 82 | 85 |
| Gap | Balanced | **10,386** | 10,143 | *10,519* | 10,213 | 10,308 | 10,323 | 10,336 | 10,358 |
|  | Bias (loss) | 628 | 799 | *435* | 726 | 659 | 630 | **600** | 571 |
|  | Bias (flux) | *112* | 146 | 150 | 160 | **137** | 143 | 139 | 142 |
|  | Bias (local) | *122* | 165 | **149** | 161 | 151 | 156 | 177 | 180 |
|  | Outliers | 22 | 17 | 17 | *10* | **15** | 18 | 18 | 19 |

is illustrated in Additional file 1: Fig. S4. BWA-MEM -L  30 generally performed somewhat better than Minimap2 in all categories.

Comparing results for the various Giraffe indexes, we observed that the number of biased sites decreased as we moved from the linear reference (Giraffe-linear) to the references inclusive of more genetic variation (major, pop5 and 1KGP), with the reduction being chiefly due to loss events. The trend holds for both SNVs and gaps. We repeated the analysis on chromosome 16, giving similar results as for chromosome 20 (Additional file 1: Table S1).

Figure 4 shows bias-by-allele-length plots including each aligner, along with the SB baseline (blue). Note that all of the balance measurements are modified to put the ALT in numerator for consistency with past studies. Panel **a** shows mapping balance (MB), and **b** shows assignment balance (AB) using the context-aware algorithm. In all cases, the lines tend to diverge more for the more extreme-length insertions and deletions. The bias noted for longer insertions seems to be greater than that of longer deletions. Note that reads carrying inserted sequence contain fewer bases that align to the reference, which in turn makes them harder to align correctly. This is in contrast to reads spanning deletions, which still align well to the reference genome, albeit with a deletion-sized gap. In addition, reads carrying insertions can sometimes not spanning the whole insertion, and have only one end overlapping the reference. BWA-MEM -L  30 stays the closest to simulation balance followed by Bowtie2, VG Giraffe, and the default settings of BWA-MEM is the most biased. Across the different Giraffe indexes, the balance improves from the linear to the major, pop5, and 1KGP indexes.

### Measuring bias using real reads on well-characterized genome

Biastools can also be applied to study reference bias in real datasets. Here we discuss biastools's usage when reads come from a well-studied individual for which we have foreknowledge of HET sites. Since simulation balance (SB) and mapping balance (MB) relied on information from the simulator, we do not use them here. We continue to use assignment balance (AB) including with the context-aware assignment algorithm.

#### *Visualizing bias*
We made the bias-by-allele-length plot shown in Fig. 4c. Since simulation balance is not available as a baseline, we used an ALT fraction of 0.5 as the baseline. The trends observed were similar to those observed for simulated data (Fig. 4a, b). BWA-MEM with the -L  30 option and Bowtie 2 had the most even balance for longer insertions and deletions. For VG Giraffe, the indexes that included more variants had less bias than the indexes with fewer variants.

#### *Classifying biased sites*
Given a set of read alignments, biastools can predict which sites were affected by reference bias. To do this, biastools first performs context-aware assignment and measures allelic balance at the HET sites. Biastools also measures the mapping quality of the alignments overlapping each HET site, since low mapping-quality reads indicate possibly mis-mapping due to repeats.

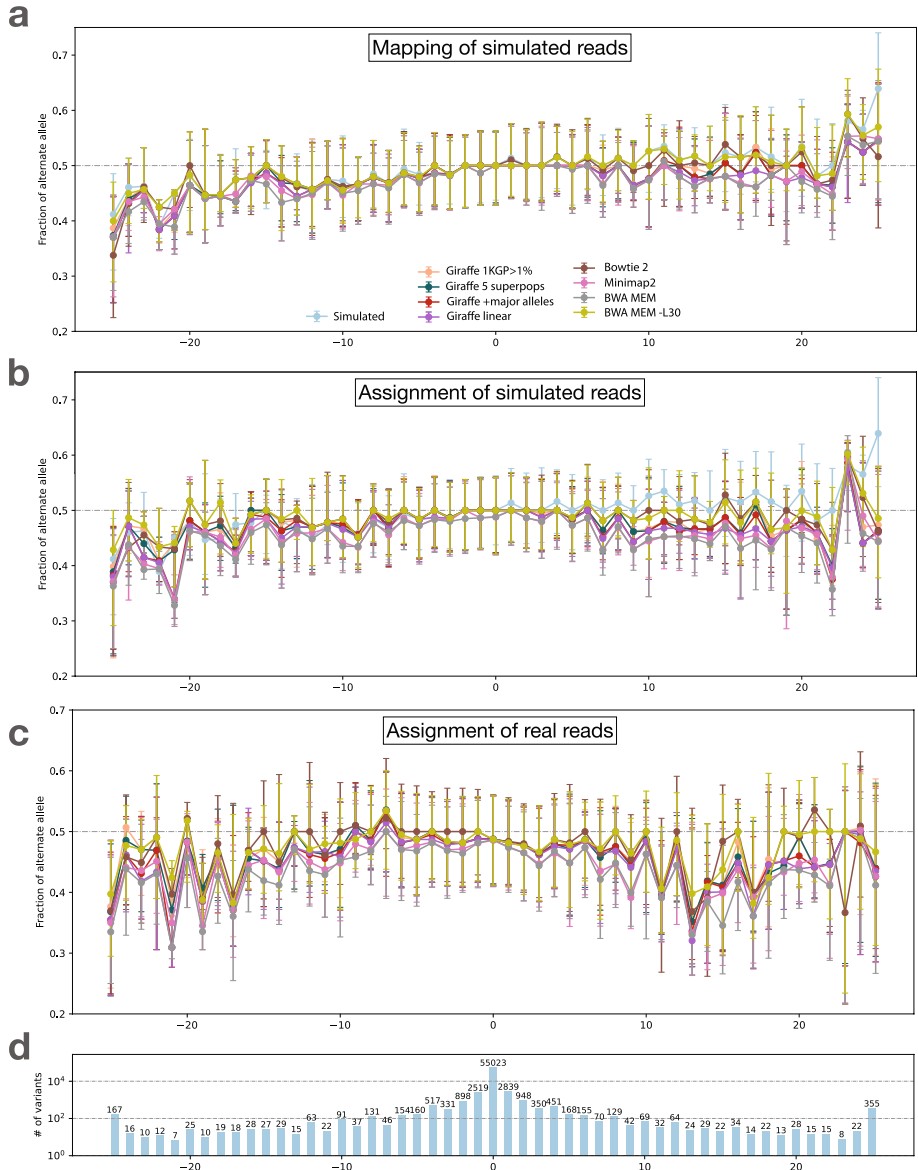

**Fig. 4** Bias-by-allele-length for 8 alignment workflows. We used simulated and real WGS datasets derived from HG002. We subsetted to reads aligning to HET sites on chromosome 20. Variants are arranged according to their length, with positive values standing for insertions and negative values standing for deletions. Zero indicates SNVs. **a** Fraction of ALT alleles in the simulation (blue) and after mapping of simulated reads (other colors). **b** Fraction of ALT alleles after mapping and context-aware assignment using simulated reads. **c** Fraction of ALT alleles after mapping and context-aware assignment using real reads. **d** The number of incidents of each size

We hypothesized that a combination of (a) allelic balance and (b) the average mapping quality of the overlapping reads could be used to predict if a variant site is affected by reference bias. We combined allelic balance (which varies from 0 to 1) with average mapping quality (normalized to vary from 0 to 1) using both addition and multiplication, then used these to rank sites according to their likelihood to be affected by bias. We applied these both to the simulated read data aligned by Bowtie 2, and to the real reads

aligned by Bowtie 2. At each HET, we applied the classifier and compared its true/false categorization to the categorization obtained using the NMB-NAB analysis detailed in the "Biased-site classification" section. Recall that the NMB-NAB categorization uses information about the simulated points of origin to classify sites as balance or as one of several bias event categories: loss, flux, local or outlier. In this evaluation, we collapse these into a single "biased" category.

While we lack ground-truth information about which HET sites are biased for the real reads, we assumed that bias events observed in the HG002 simulation would also occur in the real HG002 reads. That is, we transferred the ground-truth bias labels from the simulation to the real data. Figure 5 shows the receiver operating characteristic (ROC) curve and precision/recall (PR) curve evaluating our two-feature classifier. Panels a and b show the resulting curves for SNVs. Panel a shows that the classifier had area-under-curve (AUC) above 0.95 in all cases, whether we used addition or multiplication to combine features, and whether we evaluated on simulated or real reads. The PR curve for SNVs (panel b) had area-under-precision-recall-curve (AUPRC) ranging from 0.87 to 0.91. Further, the PR curves showed a more pronounced difference whereby classification accuracy for real data was lower than for simulated data.

For gaps, however, the ROC (Fig. 5c) and PR (Fig. 5d) curves were noticeably worse than for SNVs, with AUC of ROC ranging from 0.83 to 0.89, and AUPRC ranging from 0.57 to 0.66. That is expected, since the majority of biased SNV sites are loss or flux events that are well characterized by our allelic balance and average mapping quality

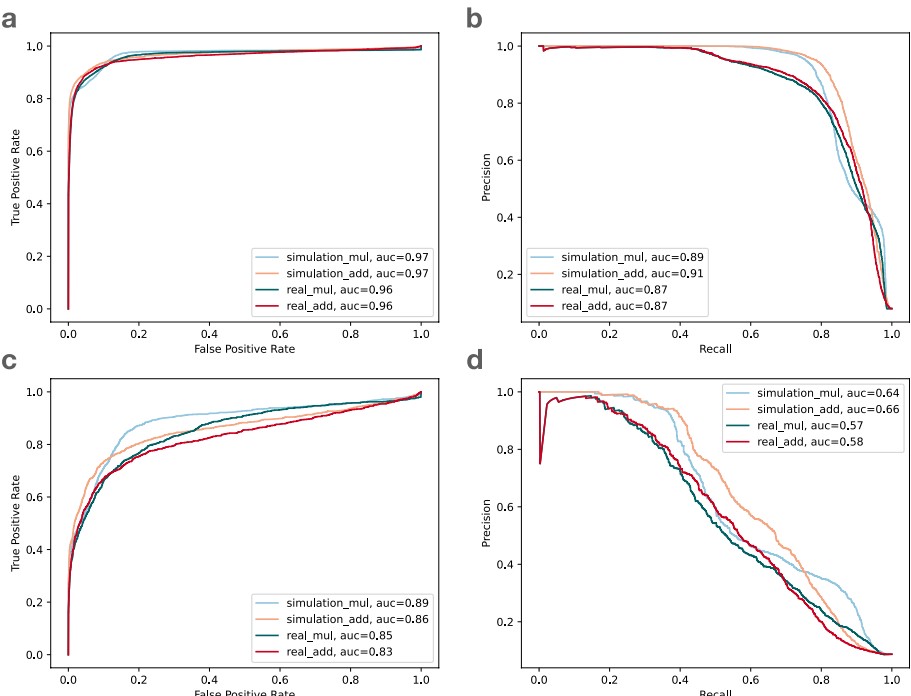

**Fig. 5** The receiver operating characteristic (ROC) curve and the precision and recall (PR) curve of the biastools classifier on Bowtie2 alignment. **a** ROC curve of SNVs, **b** PR curve of SNVs, **c** ROC curve of gaps, **d** PR curve of gaps. The four lines are the simulated (blue and orange) and real data (green and red) based on multiplication scoring (mul) and addition scoring (add). auc: area under curve

features. For gaps, however, a larger proportion of the bias comes from loss or local events, and our features are only partially effective at capturing local bias.

**Measuring bias using real reads from an uncharacterized genome**

While the above experiments used either simulated reads or foreknowledge of HET sites, a common scenario is that the reads come from donor individual with unknown variants. We hypothesized that `biastools` could still detect biased regions based on three measures: (a) read depth, (b) density of ALT alleles detected, and (c) frequency of sites for which the evidence is inconsistent with a diploid state. We expect some or all of these measures to become extreme in areas affected by reference bias. For example, if a donor has multiple copies of a segmental duplication that exists in a single copy in GRCh38, reads from the duplicates will accumulate in a single region on GRCh38, leading to higher depth and, due to the collapsed evidence, some non-diploid variants.

`Biastools`'s scan mode computes windowed running statistics over the pileup. In each window, it computes a read depth (RD) score, variant density (VD) score, and non-diploidy (ND) score, each of which are ultimately transformed to $Z$ scores. The $Z$ scores are then combined by taking their sum. Regions with combined score $\geq 5$ are called "biased" and regions with score in the interval $[3, 5]$ are called "suspicious." When biased regions are close to each other (within 1 kbp), they are combined to make one longer biased region. This combining also happens for suspicious reasons. Details are in the Methods "Sliding window approach of scan mode" section.

To evaluate `scan` mode, we ran it on the simulated HG002 read data from "Measuring sources of bias in simulation" section aligned by Bowtie 2 to the full GRCh38 reference. It reported 72,165 biased regions of average length 872 and 90,368 suspicious regions of average length 326 across the genome. While the input to this experiment was simulated data, our analysis does not use any information from the simulater, nor does it use foreknowledge of HG002's variants. Focusing on chromosome 20, we compared the regions called biased by `scan` mode with the variant sites that were called biased using context-aware assignment as described in "Results" "Measuring sources of bias in simulation" section. `scan` mode called 3384 biased regions on chromosome 20, covering 4.9% of its bases. Of the SNVs and gaps on chromosome 20 that the `biastools` classifier (which does use simulation information and foreknowledge of HETs) calls balanced, 81% and 74%, respectively fell outside of the regions called biased by `scan` mode. On the other hand, 75% of SNV sites and 78% of gap sites called biased by the NMB-NAB analysis fell inside regions called biased by `scan` mode (Table 2).

In this way, `scan` mode reproduced the results of the per-site classifier in part, but not completely. This is expected since `scan` mode lacks foreknowledge of HET locations.

*Bias near structural differences*

As a further demonstration of `scan` mode, we ran it on the real HG002 Bowtie 2 alignments used in "Results" "Measuring bias using real reads on well-characterized genome" section. `Biastools scan` marked 4.6% of the GRCh38 primary assembly (considering the 22 autosomes and the two sex chromosomes) as belonging to biased regions. Since bias can be caused by missing or incorrectly collapsed sequence in the

**Table 2** Comparison of the regions found using `biastools scan` mode versus the bias sites detected using `simulate` mode. The analysis is on chromosome 20, where 3, 384 segments are called as biased by `scan` mode, adding to 4.9% of chromosome 20

|     |              | Total sites | Inside bias region | Outside bias region |
|-----|--------------|-------------|--------------------|---------------------|
| SNV | Balanced     | 52,173      | 9915 (19%)         | 42,258 (81%)        |
|     | Bias (loss)  | 2108        | 1520 (72%)         | 588 (28%)           |
|     | Bias (flux)  | 495         | 418 (84%)          | 77 (16%)            |
|     | Bias (local) | 169         | 142 (84%)          | 27 (16%)            |
|     | Outliers     | 78          | 61 (78%)           | 17 (22%)            |
|     | Bias (all)   | 2850        | 2141 (75%)         | 709 (25%)           |
| Gap | Balanced     | 10,386      | 2707 (26%)         | 7679 (74%)          |
|     | Bias (loss)  | 628         | 482 (77%)          | 146 (23%)           |
|     | Bias (flux)  | 112         | 102 (91%)          | 10 (9%)             |
|     | Bias (local) | 122         | 81 (66%)           | 41 (34%)            |
|     | Outliers     | 22          | 22 (100%)          | 0 (0%)              |
|     | Bias (all)   | 884         | 687 (78%)          | 197 (22%)           |

**Table 3** Number of structural variants (SVs) longer than 100 bp called by HGSVC and falling either within or outside the biased regions identified by `biastools scan`. An SV was called as within the region of it overlapped any position within 100 nt of the extends of the SV. The biased regions consisted of 114,845 total segments, adding to 4.1% of the length of the GRCh38 primary assembly

|           | SV number | Inside bias region | Outside bias region |
|-----------|-----------|--------------------|---------------------|
| Insertion | 9709      | 6183 (64%)         | 3526 (36%)          |
| Deletion  | 5521      | 2690 (49%)         | 2831 (51%)          |
| Total     | 15,230    | 8873 (58%)         | 6357 (42%)          |

reference, we hypothesized that the biased regions would have a tendency to be in or near HG002's structural variants (SVs). We examined HG002's SVs as called by Human Genome Structural Variation Consortium (HGSVC) [12] with length over 100 bp, finding 9709 insertions and 5521 deletions. We found that 6183 insertions (64%) and 2690 deletions (49%) fell inside or within 100 bp of regions called biased by `scan` mode (Table 3). The greater enrichment of insertions in biased regions was expected, since reads containing inserted non-reference sequence are more likely to align incorrectly.

### Bias due to incomplete reference representations

 We used `scan` mode to compare two different reference representations and alignment strategies. The first used Bowtie 2 to align directly to GRCh38, as we did above. We call this the "direct-to-GRC" method. The second used a workflow that additionally makes use of the complete telomere-to-telomere (T2T) CHM13 human genome assembly. The second method uses Bowtie 2 to align first to the more complete T2T-CHM13 assembly. Then, for reads that fail to align unambiguously to T2T-CHM13, it additionally aligns those to the GRCh38 assembly. For reads that align successfully to both, the alignment with the higher alignment score (to its original target, not necessarily to GRCh38) is chosen. After merging, all alignments are ultimately "lifted" to GRCh38, i.e., translated into GRCh38 coordinates. We call the second method — the one that uses both T2T-CHM13

and GRCh38 — the "LevioSAM 2" method, since it was first proposed in the LevioSAM 2 study by Chen et al. [5].

We found that 4.0% of the LevioSAM 2 alignments fell into regions that were classified as biased by `biastools scan` (125,498 biased regions, average length 974 bp), compared to 4.5% for the direct-to-GRC alignments (130,771 regions, average length 1071 bp). We used `bedtools subtract` to find the regions called biased using one method (direct-to-GRC or LevioSAM 2) but not the other. Out of the 130,771 regions in direct-to-GRC, 27,831 (21%) were improved by more than 25% bases when using LevioSAM2. In contrast, 11,447 (9%) bias regions in "LevioSAM 2" were aligned more balance in "direct-to-GRC".

Since the improved performance of the LevioSAM 2 workflow is related to the completeness of the T2T-CHM13 reference relative to GRCh38, we hypothesized that the improvements would tend to be in regions where the T2T-CHM13 assembly is known to be superior, such as centromeres. We define that a bias region is near the centromere if it is inside the centromeric region or within 500*k* range extend from the centromere. The summation of the extended centromeric regions contains 86,076,358 bp, which is around 3% of the whole genome. We collect all the bias regions improved by 25% in LevioSAM 2, and measure how many improved bases are from the regions near centromere, and how many are not. Thirty-eight percent of the improved bases are actually near centromeres, which is a high enrichment comparing to 3% (Table 4). Furthermore, if we consider only the bias region greater than 1000 bp, the ratio of bases near centromere becomes 40%, indicating that the bias region near the centromere tends to be longer.

Figure 6 illustrates a region near a centromere where the direct-to-GRC method yields more reference bias compared to LevioSAM 2. Non-gray colors (blue, red, green, orange) in the IGV pileup denote places where alignments carried an ALT allele relative to GRCh38. The top pileup shows that direct-to-GRC alignment created a dense area of ALT alleles (evident from the density of non-gray coloring). Further, the direct-to-GRC alignments tended to cover the region to much higher depth compared to the LevioSAM 2 alignments, evident from the scaling of the top (0–254) and bottom (0–60) coverage tracks. These factors indicate that, for direct-to-GRC alignment, reads from more than one region of the donor genome have aligned in a "collapsed" fashion to this single region, create extreme values for RD, VD and ND and causing `biastools`'s `scan` mode to mark the entire region as biased.

The LevioSAM 2 pileup exhibits much less bias, though `biastools`'s `scan` mode reports some small biased regions here, as can be seen in the bottommost panel. The contrast between the combined RD, VD and ND score is illustrated toward the top of

**Table 4** The location of the regions improved by 25% reference bias when using LevioSAM 2 compared to the direct-to-GRC method. A biased region was "Near Centromere" if it locates within 500 kbp of a region annotated as centromeric by the UCSC genome browser

| bases improved | Total | Near centromere | Away from centromere |
|---|---|---|---|
| All regions | 6,132,182 | 2,304,633 (38%) | 3,827,549 (62%) |
| Regions > 1000 bp | 5,631,958 | 2,242,093 (40%) | 3,389,865 (60%) |

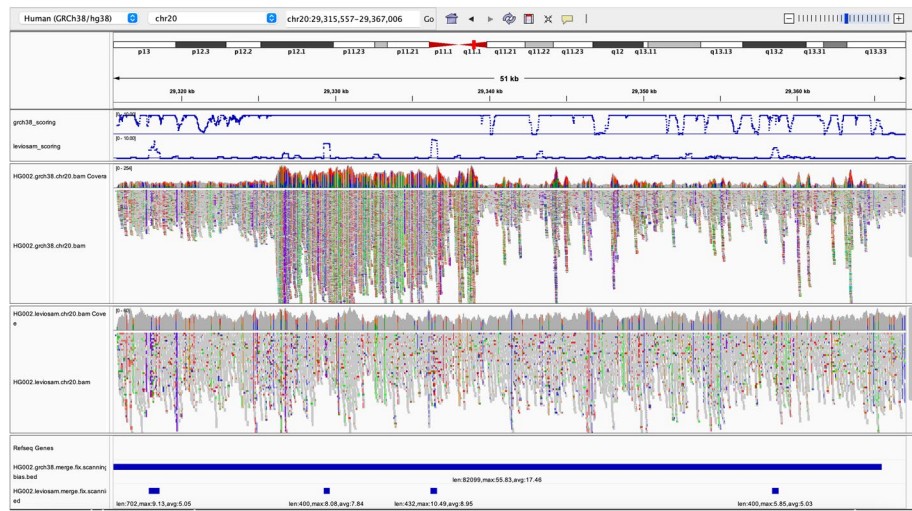

**Fig. 6** Biastools called bias region of HG002 with two different method. The tracks from top down are: combined *Z*-score for direct-to-GRC alignment, combined *Z*-score for Leviosam2 alignment, IGV read arrangement of direct alignment, read arrangement of Leviosam2, "Biased region" of direct alignment, "Biased region" of Leviosam2. Combined *Z* scores include read depth, variant density, and non-diploid variant. The scores above 10 are truncated in the panel to show the details between 0 and 10. Note that the read coverage tracks use different scales. For direct alignment, the track ranges from 0 to 254, while that of Leviosam2 ranges from 0 to 60

**Table 5** Time and memory usage for different stage of biastools when processing whole genome simulated data. The alignment method is Minimap2, and the sorting is of the alignment is done by `Samtools sort`. All the experiment are done with single thread

|  | Time (hr) | Memory (GB) |
| --- | --- | --- |
| Alignment and sorting | 21.73 | 12.32 |
| Biastools assignment (sim) | 9.25 | 37.90 |
| Biastools assignment (real) | 8.46 | 15.39 |
| Biastools scan | 47.38 | 368.17 |
| Biastools scan from mpileup | 6.42 | 368.07 |

the screenshot, where the blue curves show the combined score, truncated to remain in the interval [0, 10]. The threshold determining biased or not is on 5. Most of the regional score in direct alignment (upper track) are actually above 10 while only a few region in Leviosam2 (lower track) reach 10.

### Computational performance

To test the computational efficiency of biastools, we performed experiments using the simulated WGS data on a Linux x86_64 system with single thread (Table 5). While the various alignment tools take different amounts of time, Minimap 2 was the fastest. As a result, we used Minimap 2, plus the necesary alignment sorting task, as the baseline for our measurements. After alignment and sorting, context-aware assignment and generation of the bias report (`sim` mode) took 9.24 h while using a peak memory footprint of 37.90 GB. When run on the real WGS reads (without ground truth information), biastools' assignment phase took 8.46 h and used a peak memory footprint of 15.39 GB.

To run `biastools_scan --scan`, the input file must be in mpileup format. Transforming `.bam` format to `.mpileup` with `bedtools`, then performing the `biastools_scan --scan` took 47.38 h, while performing `biastools_scan --scan` on an existing mpileup file took 6.42 h. The peak memory usage in either case was around 368 GB.

## Discussion

We presented `biastools`, a novel method and tool that directly measures and categorizes instances of reference bias. In a simulation setting, we demonstrated its utility for identifying different categories of reference-bias events, and used this facility for comparing some well known alignment methods. Using real data, we showed its accuracy in a range of situations, including when we either do or don't have foreknowledge of the donor individual's HET sites.

As the bioinformatics community continues to develop new bias-avoiding methods [15] we expect `biastools`'s ability to measure and categorize bias events will be essential. Direct measurement of reference bias will lead to clearer interpretation and evaluation compared to the alternative of measuring accuracy in a downstream result like variant calling. Findings obtained using `biastools` will help in designing the next generation of reference representations and alignment algorithms. For instance, our finding that end-to-end alignment leads to less bias in some circumstances could indicate that future algorithms should favor end-to-end alignments in more situations.

By measuring reference bias at an early point in the alignment process, `biastools` can disentangle reference bias due to the aligner and reference representation from any bias caused by downstream tools. This is particularly important since downstream tools can themselves be tuned (or trained) to counteract reference bias, sometimes "learning" the bias, when the more effective measure would be to analyze and remove the bias upstream. An example is the DeepVariant variant caller, which can refuse to call variants in bias-prone regions of the genome [5].

In the future, it will be important to refine `biastools`'s models for predicting whether a given site is experiencing reference bias. In particular, the model presented here in "Results" "Measuring bias using real reads on well-characterized genome" section performs well for relatively simple variants like SNVs, but not as well for gaps. To improve the utility of `biastools`, it will be important to include more information in this model to allow for more accurate predictions. In particular, a future task is to develop models that both identify relevant features (beyond coverage and MAPQ) and combine to make a prediction in an automated way, possibly using deep learning. Indeed, such models may exist within the larger models already developed for variant calling in tools like DeepVariant [29]. To date and to our knowledge, no existing model is designed for the specific task of measuring reference bias, which is key to understanding how well upstream tools are fulfilling their stated purpose.

Currently, biastools supports only diploid genomes, since most of the work on reference bias avoidance has focused on human and other diploid genomes. However, biastools in principle can be extended to genome with higher ploidy. For instance, the simulation and the assignment methods would be essentially the same for a triploid, with the expected allelic balance ratios being 1:2 or 1:1:1. Note that the problem

of distinguishing reference bias from sequencing error becomes harder as the ploidy increases.

This study focuses on short reads, since their shorter length makes them more prone to reference bias. However, `biastools`'s methods are applicable to long-read alignments as well. Reference bias will manifest differently for long reads compared to short ones. Since long-read aligners have the benefit of longer sequence length and more anchors, scattered pockets of dense ALT alleles are less likely to affect the aligner's ability to place the read correctly. In light of this, we expect `biastools`'s scan mode to be particularly well suited to identifying the larger-scale bias events that are likely to dominate the reference bias landscape for long reads.

## Conclusions

`Biastools` is a novel method and tool that directly measures and categorizes instances of reference bias. As new reference representations and alignment tools continue to be developed, `biastools` can help to standardize and formally measure the degree to which they address the reference-bias problem.

## Methods

### Biastools workflow

`Biastools` analyzes, measures and reports instances of reference bias in short-read alignments. `Biastools` focuses on bias with respect to diploid genomes, though the constituent methods could be generalized to other ploidies. If genetic variants are not known for the donor genome, `biastools`'s scan mode reports regions that are "biased" or "suspicious." If the donor has known variants, `biastools`'s predict mode performs a more detailed analysis, taking bias measurements at each heterozygous site. `Biastools`'s simulate goes a step further by first running a read simulator, then analyzing the simulated reads with one or more read alignment workflows. This allows for detailed categorization of bias events (e.g., whether they are due to loss, flux, etc), and for comparative studies of bias caused by different tools and reference representations.

### `simulate` *mode*

To obtain a diploid reference from which reads can be simulated, `biastools --simulate` first uses `bcftools consensus` to generate the two FASTA-format haplotypes for the donor individual from a reference genome and a set of phased variant calls in VCF format. `biastools --simulate` then uses `mason2` to simulate Illumina-like short reads from the autosomes of the two haplotypes. `biastools --simulate` uses different random seeds for the two haplotypes, to avoid correlation between the read coverage profiles. Note that `mason2` annotates simulated reads with their haplotype and point of origin. In our experiments of the "Results" "Measuring sources of bias in simulation" and "Measuring bias across aligners" sections, the individual with high quality variant calls was HG002, the VCF file used was from the Q100 project. The VCF provides the phased variant information of HG002. We filtered out the variants that had been placed in any "FILTER" category, including variants that lacked evidence on one haplotype.

Simulated reads are then aligned to the GRCh38 primary assembly with one or more user-specified read alignment workflows. Bowtie 2 and BWA-MEM align directly to an index of GRCh38. VG Giraffe aligns to a graph based on GRCh38, and with all read alignments ultimately surjected ("lifted") onto GRCh38. For each variant site, `biastools` analyzes the site using both its naive and its context-aware assignment methods, detailed in the "Methods" "Assignment method" section. Given the evidence supporting the REF and ALT alleles, three levels of allelic balance are calculated: the simulated balance (SB), mapping balance (MB), and assigned balance (AB). SB and MB require information about the reads' true haplotype and point of origin, which are provided by the simulator, whereas AB is based only on the results of the context-aware assignment assignment method ("Methods" "Assignment method" section). These measures in turn allow `biastools` to categorize HET sites, as detailed in the "Methods" "Biased-site classification" section.

### `predict` *mode*

This mode, `biastools --predict`, uses its context-aware assignment method to analyze each variant site. Since we lack simulated ground truth, only the AB measure is computed. This is sufficient to predict instances of reference bias (see the "Results" "Measuring bias using real reads on well-characterized genome" section), and to create diagnostic plots like the bias-by-allele-length plot (Figs. 3 and 4).

As presented in the "Results" "Measuring bias using real reads on well-characterized genome" section, `biastools` can predict which HET sites are affected by reference bias using a simple model. The model uses two inputs computed by `biastools --predict`: (a) the average mapping quality (MAPQ) of all the reads overlapping the site, and (b) the allelic balance at the variant site. This model is too simplistic to divide instances of bias into categories such as flux and loss. Still, our evaluations of the simple model, using simulated data to obtain ground truth for testing, indicates that it performs quite well on data derived from HG002 and aligned to GRCh38.

### `scan` *mode*

`biastools_scan --scan` first uses `samtools mpileup` to transform the alignments into the column-wise mpileup format. `Biastools` then scans the mpileup file, performing a windowed analysis and seeking regions with unusual degrees of (a) depth of coverage, (b) SNV variant density or (c) instances where the evidence is inconsistent with a diploid donor genome. The three measurements are combined into a single score by adding or multiplying them. Regions having combined score above a threshold are marked as "biased." We cross check the scanning mode with both simulated data and real data.

### Assignment method

`Biastools` contains two algorithms (the "naive" and the "context-aware" algorithms) for assigning reads to haplotypes. Both examine each read that aligns across a given site and assign each read to the reference-allele-carrying (REF) or the alternate-allele-carrying (ALT) haplotype. This problem is made difficult by the presence of sequencing errors, ambiguity in placement of alignment gaps, and the presence of repetitive sequence.

While both algorithms attempt to assign each read to one haplotype or the other, they can fail in the case of some reads, ultimately assigning them to neither haplotype.

Before describing these assignment methods, we first describe how `biastools` computes two different baselines for understanding allelic balance.

### Simulated balance (SB)

SB is computed as the number of ground-truth REF reads simulated from across the site, divided by the total number of reads simulated across the site. That is, it is the ratio REF/(REF+ALT), where REF and ALT are obtained by examining the simulated reads and simply counting the number that overlap the site and come from the REF-carrying haplotype and ALT-carrying haplotype.

### Mapping balance (MB)

MB is computed as the fraction of reads overlapping the site that both (a) originated overlapping the site, and (b) aligned overlapping the site. Information from the read simulator is used to determine the read's haplotype and point of origin. Reads that aligned overlapping the site but that were actually simulated from elsewhere in the genome are not counted in the MB measure. The MB measure differs from the SB measure since some reads truly originating from the site will fail to align there.

VCF files can contain nearby variants that are interdependent in a way that prevents the sites from varying independently. For example, a deletion could extend through and cover an SNV; that is, the deletion removes the SNV site, making the SNV neither REF nor ALT. Some VCF files use "./." to represent such cases. To avoid the complications that arise from these cases, we identified instance of overlapping variants and removed them from consideration by ignoring all of the polymorphisms involved.

### Naive assignment method

Given all of the reads that aligned overlapping a given site, the naive assignment method examines which base(s) from the reads align to the variant's exact reference coordinates. From those, it tallies the REF/(REF+ALT) fraction. For insertions and deletions, the method only tallies a read if its sequence exactly matches the ALT or REF allele. If the sequence is different from both reference and alternative allele, e.g., if the sequence was affected by one or more sequencing errors or if the placement of gaps or insertions was different from the VCF, the read is classified as "other" and is not counted.

Note that this method uses the exact base-by-base alignment information reported by the read aligner. In other words, decisions made by this assignment method are essentially the same as those that would be made by examining the pileup columns corresponding to the variant. The following context-aware method improves upon the naive method by reanalyzing the read sequences.

### Context-aware assignment method

This method works by searching for the REF and ALT alleles, together with some of their flanking sequence, within the sequences of all the reads that aligned overlapping the variant. As a first step, this method extracts variant information from the VCF, constructing strings that represent the REF/ALT alleles together with their flanking sequence. We

use the term "allelic context sequence" to describe the allele together with its flanking sequence." The default flanking sequence length is 5 bp. Note that flanking sequences are drawn from the same haplotype as the allele; e.g., if two phased SNV variants are within 5 of each other, each will appear – phased appropriately – in the other's flanking sequence.

To determine if a read overlapping a variant site supports the REF or ALT allele, the read sequence is scanned for the allelic context sequences for REF and/or ALT. If exactly one of the two (REF or ALT) context sequences is found, the read is classified accordingly. The allelic context sequence need not appear in its entirely; it is sufficient for a suffix or prefix to appear, as long as a suffix or prefix of the other does not also appear. A read may contain context sequences but lack the context to distinguish REF from ALT. That is, the read sequence may contain equally good matches for both alleles. This is particularly common in regions with tandem repeats. In this case, the read is classified as "both" REF and ALT for the purpose of tallying bias. In cases where the read sequence contains neither of the allelic sequence contexts, the read is classified as "other". "Both" and "other" reads are excluded from the AB calculation.

Subtleties can arise when many variants are clustered close together, with some variants (i.e., indels) affecting the coordinates at which others occur. In such cases, the evidence for any one of the variants is best understood in the context of the entire phased cluster of variants. This type of method has been adopted by multiple previous tools when analyzing variant combinations that might involve indels [2, 36]. The context-aware assignment method will cluster variants appearing within a short distance (default: 25 bp) together into a "cohort." The cohort extends in either direction until no other variants can be reached (up to the distance) in either direction. For such variants, the context-aware assignment algorithm will first take the entire (clustered) REF and ALT strings and search for them within the sequences of the overlapping reads. A read assigned in this way is tallied with respect to all of the variant sites making up the cohort. That is, if three phased variants are involved in a cohort, and the REF allele string is found in a read, that read counts toward the REF tally for all three variants.

While some overlapping reads can be tallied in this way, some overlapping reads might not overlap all or much of the cohort. For reads that cannot be assigned using the entire cohort string, the assignment algorithm falls back on the variant-by-variant strategy described previously.

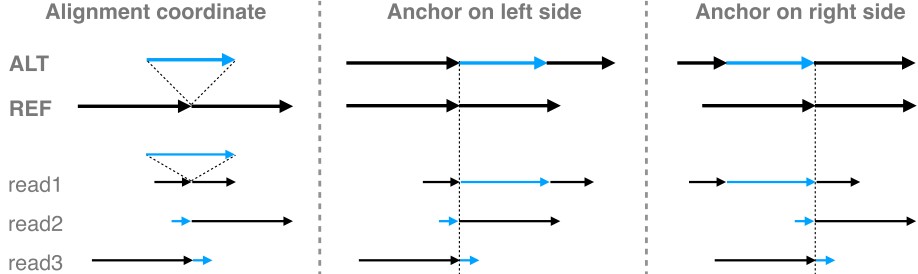

**Fig. 7** The aligned reads and variants in alignment coordinate and expansion coordinate. For expansion coordinate, the expansion can be anchored on the left side of the variant or the right side of the variant

When comparing the context sequences to the REF and ALT sequences, there may be a need to try different anchoring points, especially when gaps and tandem repeats are present. This is illustrated in Fig. 7. On the left, the coordinate of read1 is the same as reference because its insertion is anchored on both sides and correctly placed. But for read2 and read3, the inserted sequence cannot be anchored on one side, causing their coordinates to shift with respect to the reference. To deal with this, the context-aware method will first try anchoring the read on the left side boundary of the variant. If no match is found between context sequence, REF and ALT, the method will try anchoring the read on the right side boundary of the variants (Fig. 7). In the same fashion, when comparing the read sequence through the cohort of a set of variants, the left and right end of the cohort are anchored to comparison. In this way, the aligner's placement of gaps does not affect the comparison as long as the alignment beyond one of the variant boundaries is correct.

Note that the context-aware comparison method has limitations in cases where the variant calling file provides only partial information. For instance, when true variants are missing from the VCF, bias measurements at nearby sites can be inaccurate because `biastools` lacks the accurate flanking sequences needed for context-aware assignment. Similarly, absence of accurate (or any) phasing information can interfere with `biastools`'s ability to establish accurate flanking sequences for assignment.

### Repetitive context

When a variant is situated in or near a tandem repeat, it may not be possible to distinguish REF and ALT alleles simply by taking a fixed sequence context. For example, in Fig. 8a, the REF haplotype contains `attc` repeated 7 times in tandem. The ALT haplotype has the same sequence repeated only 6 times. If we only compare the variant region defined in the VCF, which is 1 `attc` difference, it is easy to mistake a read with one `attc` deletion to reference read if the aligner didn't put the gap in the exact place.

To cope with the complication, we defined the concept of "effective variant". When building the variant map, if one context sequence (REF or ALT) is the prefix, suffix, or substring of the other, the "context-aware" method will keep extending the variant. If

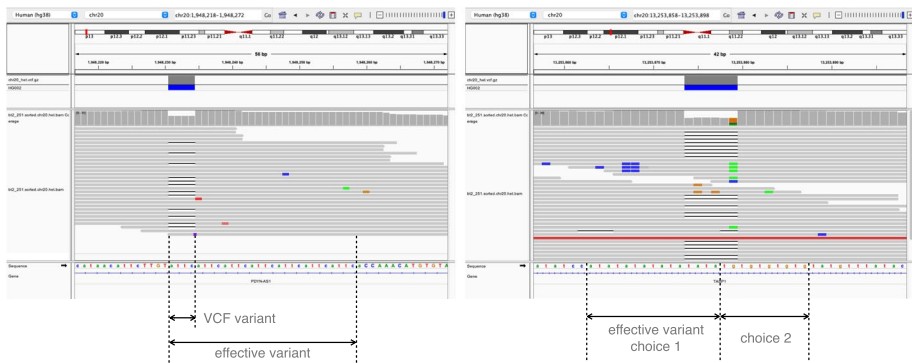

**Fig. 8** Two examples of repetitive context. **a** The repetitive is extending to the right side, so the effective variant is extending to the right end so that the ALT context sequence is no longer a prefix of REF context sequence. **b** The case original ALT context sequence is a substring of REF context sequence. There are two choices of effective variant. `Biastools` would chose the shorter effective variant (choice 2)

the repetitiveness is on the right side, that is, one context sequence is the other's prefix, the method will extend the variant to the right until the first difference is encountered. For example, in Fig. 8a, the effective variant become the whole repetitive region of the `attcs`. Similarly, the method will extend the variant to the left if the repetitiveness is on the left, that is, one context sequence is the other's suffix. Occasionally, the repetitiveness is on both side (Fig. 8b). In these cases, the method chooses the side where the extension is shorter. A read that does not cover the entire effective variant will be classified as "both," reflecting the fact that we cannot determine the true origin of a read that does not cover the whole repetitive region. Reads that partially cover the effective variant are not evaluated in our simulation experiment, since they are not possible for the assignment method to determine the haplotype and only complicated the result when being included in analysis. The variant with "effective variant" longer than 70 bp are also disregarded in analysis.

### Biased-site classification

For simulated reads, we can diagnose the cause of the bias by examining our bias measures (AB from the two assignment methods) as well as our baseline measures (SB and MB). We divide biased sites into three categories (or "events"): loss, flux, and local. Loss events are caused by ALT-carrying reads that fail to align to their true point of origin. Flux events are caused by reads that aligned to a site but that originated from another site on the genome. Local events are caused by the aligner put the reads in roughly correct place; however, the reads' haplotype is determined incorrectly by the assignment method. It can be due to the assignment method is fooled by the placement of the gaps such as the most "local" bias cases in naive assignment. The "local" bias also happens when the aligner put the read off certain bases due to the tandem repeats or the uneven incidents of sequencing error in the variant region.

We rely on three combined measures to classify the biases. One is the normalized mapping balance (NMB), equal to MB - SB. NMB measures bias that manifests due to read alignment. Another combined measure is the normalized assignment balance (NAB), equal to AB - SB. NAB measures bias that manifests due to either read alignment or a failure to correctly tally the evidence present in the overlapping aligned reads, e.g., due to ambiguity caused by gaps and tandem repeats. A final measure is the number of reads that aligned to the site incorrectly due to having ambiguous alignments.

Our bias categories are defiend based on these three measures. Figure 9 illustrates how categories are determined based on the measures. Most sites generally do not exhibit reference bias, and so would tend to appear near the origin of the plot, meaning that MB and AB are both close to SB. Specifically, any sites falling within the circle about the origin with radius 0.1 are classified as "balanced."

The yellow region that surrounds the diagonal $y = x$ line in Fig. 9 (but excluding the "balanced" circle around the origin) demarcates the sites that are categorized as "loss" events. The boundary is specifically defined by two lines with slopes of 2 and 1/2. Positioned along the diagonal means that the NMB and NAB are close to each other, indicating that the assignment method reflects the balance of reads mapping to the site. However, positioning in the upper-right quadrant means that these sites are biased toward the reference, which results from the loss of alternative allele reads. In some rare

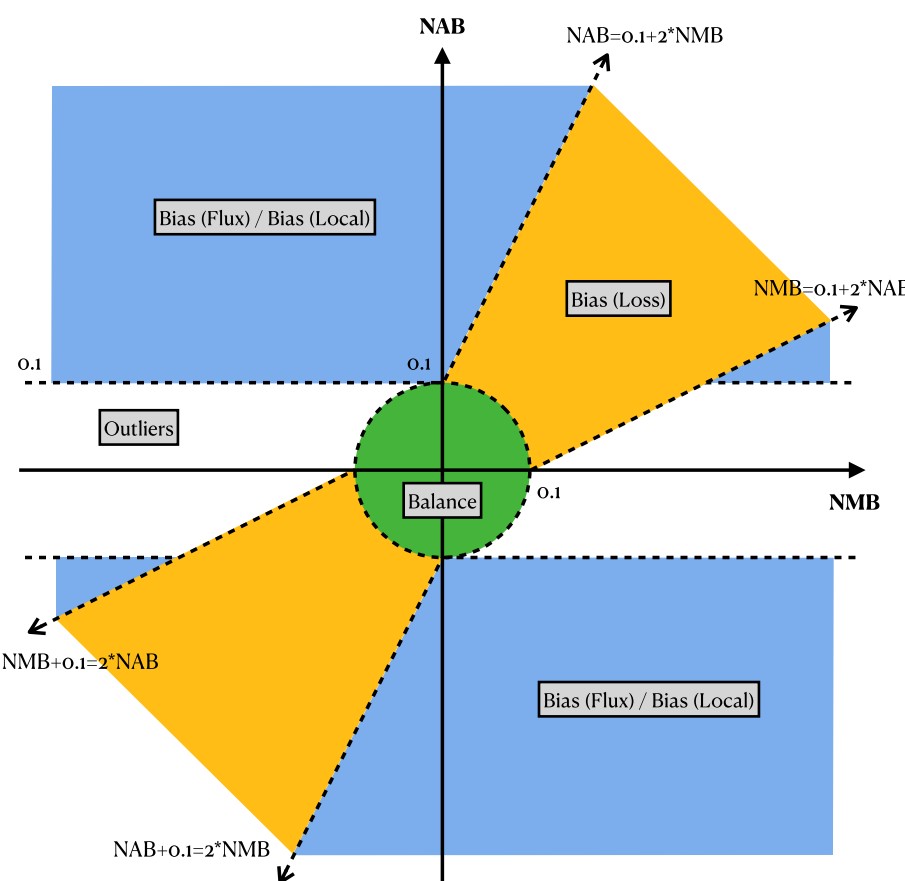

**Fig. 9** The illustration of biases categorization with NAB and NMB. Variants positioned within the green circle with a radius of 0.1 at the origin are classified as balance. Variants in the yellow region along the diagonal are categorized as bias "loss". The blue region, where |NMB| > 0.1 and excluding the bias "loss" region classifies variants as either bias "flux" or bias "local". The classification between "flux" or "local" is determined by if there are more than 5 reads being mismapped to the site. Variants falling outside these categories are classified as outliers. NAB: normalized assignment balance, NMB: normalized mapping balance

case the reads carried reference allele are lost, in the case the variant site will situated in the lower-left quadrant. The blue region in Fig. 9 are where the variants with discordant NMB and NAB located. In most of the cases, the NMB are close to zero, while the NAB is positive, meaning that the mapping of the reads are close to the simulation, but the assignment is not correct. Both the flux and local biases position in similar place, thus we introduce a third measure, number of mismapped reads, to differentiate these two categories. For the variant site in the blue region, if there are more than 5 reads coming from other place of the genome, then the site is classified as bias "flux", else it is classified as "local". The sites not included in the green, yellow, or blue regions are classified as outliers.

### Construction of pangenome graphs

To construct the pangenome graph, we used `vg autoindex --workflow giraffe` with the GRCh38 reference and the target VCF file. Then we used `vg giraffe` with the

index files to align both the simulated and real reads. The option `-o BAM` was used to project the alignment result back to the linear reference GRCh38.

The command we used to filter the 1KGP variants with allele frequency greater than 0.01 was `bcftools view --min-af 0.01`. The command leaves only the non-reference alleles with frequency greater than 0.01 in the population.

### Evaluating biased-site predictions

As mentioned in Results, the absence of phasing information in the "truth" VCF can create problems for `biastools`'s algorithms. Before evaluating the performance of the prediction model on real read data, we first filtered out the sites potentially being affected by incomplete phasing information. To identify these, we classified each HET site as "affected" if more than 90% of reads covering the site contained an "other" (i.e., neither REF or ALT) allele, or if evidence for one of the two HET alleles was completely absent and more than 40% of reads contained an "other" allele. We then omitted the affected HETs from further analysis.

Since these real reads have no known point of origin, the measures previously used — e.g., NMB, NAB, and number of mismapped reads — are not available. We can still evaluate AB for each variant using `biastools`'s assignment methods. We found that the most relevant measures for real read alignment are the average read mapping quality and the AB of the variant. Mapping quality of the reads is the proxy to tell if there are reads from other place align to the site, or if there are reads origin in the site align to other place. AB captures whether a variant suffers from biased read loss/gain. AB does not capture the reason for the bias; i.e., sites with unbalanced REF-ALT ratio can result from random sequencing error or systematic bias. Still, we found that variant sites with extreme AB and low average mapping quality were likely to be biased sites.

We found that transforming AB and average mapping quality into *Z* scores and combining them provided a useful measure of bias. We used two methods to combine the *Z* scores; multiplication and addition.

$$\text{prediction\_score} = \frac{\text{avg\_MapQ} - 42}{42} \times \text{assigned\_balance} \tag{1}$$

$$\text{prediction\_score} = \frac{\text{avg\_MapQ} - 42}{42} \times 1.5 + \text{assigned\_balance} \tag{2}$$

Note that 42 is the maximum possible score for Bowtie2 and BWA MEM aligner. For VG Giraffe, the maximum scoring was adjusted to its maximum of 60. We observed that these two combinations performed similarly when predicting bias of SNV variants (Fig. 5).

### Sliding window approach of `scan` mode

In `scan` mode, `biastools` uses `bcftools mpileup` to obtain an alignment pileup in the target region. `Biastools` scans the region with a sliding window (default: 400 bp), finding windowed averages for three measures: read depth, variant density, instances of non-diploid pileups. The three measures are combined into a bias score as below:

- The read depth (RD) as a *Z* score: (window mean RD - total mean RD)/(total RD std). Since we are interested only in cases where RD is much greater than average, any *Z* score less than 1 is rounded to 0.
- The variant density (VD) as a *Z* score (window mean VD - total mean VD)/(total VD std). Since we are interested only in cases where density is greater than average, negative *Z* scores are truncated to 0.
- The non-diploid (ND) score as a *Z* score (window mean ND - total mean ND)/(total ND std). Since we are interested only in cases where evidence is inconsistent with the diploid state, negative *Z* scores are truncated to 0

The non-diploid (ND) score is calculated from the ratio of nucleotides appearing in each individual position in the window. For a given position, any nucleotide appearing with greater than 15% frequency is considered as an allele (i.e., is not likely to be a coincidence of sequencing errors). Any position with more than one allele is considered a SNV. A site is called non-diploid if either (a) more than two alleles are present at the $> 15\%$ level, or if the most frequent allele has a frequency more than twice that of the second most frequent.

A region is classified as "biased" if the sum of RD, VD, ND score $\geq 5$. A region is classified as "suspicious" if the sum of RD, VD, and ND score $\geq 3$ and $< 5$. If two nearby bias regions are within distance of 1 kbp, the scanning mode will chain them into one single long biased region. In a similar fashion, sites with unusual high measures but not extremely high would be classified as suspicious sites and linked together if they are within 1 kbp range.

Note that the transformation to *Z* scores requires that `biastools` determine (or estimate) the scores' means and standard deviations in the dataset. The user can chose to have these computed automatically using a sampling method, which by default samples 1/1000$^{th}$ of genome sequence and estimates based on the alignment data in that subset. Alternately, the user can specify pre-calculated means and standard deviations.

The final score of the sliding window is:

$$\text{Z\_score(score)} = \frac{\text{window\_avg\_score} - \text{sampled\_avg\_score}}{\text{sampled\_std\_score}} \tag{3}$$

$$\text{bias\_score} = \text{trunc}(\text{Z\_score(RD)}, 1) + \text{trunc}(\text{Z\_score(VD)}, 0) + \text{trunc}(\text{Z\_score(ND)}, 0) \tag{4}$$

### Comparing two alignment workflows with `scan` *mode*

To compare alignments from two alignment workflows, we first obtained a single set of average and standard-deviation parameters, derived jointly from the alignments generated by both workflows. We found that using independently sampled parameters, i.e., obtaining separate average and standard-deviation parameters for each workflow, would create an imbalance. For example, since "LevioSAM 2" produced an overall less biased set of alignments in our experiment, the average and std values of RD, VD, and ND were lower. The biased and suspicious regions reported by `biastools scan` were therefore less extreme for "LevioSAM 2" than for the more biased workflow that aligned directly to GRCh38.

To obtain these joint parameters, `biastools scan` samples from both alignment bam files, creating a sample drawn half from one workflow and half from the other. So the `scan mode` can be performed on both bam files with the same scoring.

When comparing the biased regions from the two alignment workflows, regions with low or no read depth were excluded, since it was difficult to interpret these as being improved by one workflow or the other. An example of a dubious "improvement" is illustrated in Additional file 1: Fig. S5. To classify a region identified as "biased" in one workflow as being "improved" by the other workflow, `biastools scan` required that at least 25% of the bases in the region be both well covered (over $1/5^{th}$ of the overall average read depth) and not classified as biased.

## Supplementary Information

---

**Additional file 1: Figure S1.** Full normalized mapping balance to normalized assignment balance (NMB-NAB) plot. **Figure S2.** Normalized mapping balance to normalized assignment balance (NMB-NAB) plot stratified by allele length. **Figure S3.** Example of local decision by Bowtie 2 and BWA MEM. **Figure S4.** Example of local decision by default BWA MEM and BWA MEM with option -L 30. **Figure S5.** An example of the low coverage result of LevioSAM 2 and direct-to-GRC methods. **Table S1.** Number of balanced sites and different categories of biased sites on chromosome 16.

**Additional file 2:** Review history.

---

### Acknowledgements
This work was carried out at the Advanced Research Computing at Hopkins (ARCH) core facility , which is supported by the National Science Foundation (NSF) grant number OAC 1920103.

### Peer review information

### Review history
The review history is available as Additional file 2.

### Authors' contributions
ML, SI, NC and BL designed the method. ML and SI wrote the software and performed the experiments. ML and BL wrote the manuscript. All authors read and approved the final manuscript.

### Funding
ML, SI, NC and BL were supported by NIH grant R01HG011392 to BL.

### Availability of data and materials
The VCF file of HG002 from the Q100 project was downloaded from the GIAB HG002 GRCh38 assembly-based small and structural variants draft benchmark sets [28] with the URL https://ftp-trace.ncbi.nlm.nih.gov/ReferenceSamples/giab/data/AshkenazimTrio/analysis/NIST_HG002_DraftBenchmark_defrabbV0.012-20231107/GRCh38_HG002-T2TQ100-V1.0_smvar.vcf.gz.
 The real short read sequencing data for HG002 was downloaded from Google brain genomics sequencing dataset for benchmarking and development [3] with the URL https://storage.googleapis.com/brain-genomics-public/research/sequencing/fastq/novaseq/wgs_pcr_free/30x/.
 The software `biastools` is available at https://github.com/maojanlin/biastools with the Zenodo DOI: 10.5281/zenodo.10819028 and https://pypi.org/project/biastools/ under the MIT license. Scripts for the experiments described in this paper are at https://github.com/maojanlin/biastools_experiment, with the Zenodo DOI 10.5281/zenodo.10818966.

## Declarations

### Ethics approval and consent to participate
Not applicable.

### Competing interests
The authors declare that they have no competing interests.

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

## 