## [**Additional file 2:** Review history. · Genome Biology]

Additional File 1: Supplementary Figures and Table

Title: Measuring, visualizing and diagnosing reference bias with biastools

Authors: Mao-Jan Lin^{1,*}, Sheila Iyer¹, Nae-Chyun Chen¹, Ben Langmead^{1,*}

Affiliations:

¹ Department of Computer Science, Johns Hopkins University

* To whom correspondence may be addressed. Email: mlin77@jhu.edu,
langmea@cs.jhu.edu

Figure S1: Normalized mapping balance to normalized assignment balance (NMB-NAB) plot of **a.** SNV sites with naive assignment method, **b.** SNV sites with context-aware assignment method, **c.** insertion and deletion sites with naive assignment method, and **d.** insertion and deletion sites with context-aware assignment method. All variants are listed in the figure.

Figure S2: Normalized mapping balance to normalized assignment balance (NMB-NAB) plot from Bowtie 2 mapping simulated reads with naive assignment method on **a.** gaps between 10 and 20 bp, **c.** gaps between 20 and 50 bp, and **e.** gaps greater or equal to 50 bp. And the same dataset assigned by context-aware method on **b.** gaps between 10 and 20 bp, **d.** gaps between 20 and 50 bp, and **f.** gaps greater or equal to 50 bp.

Figure S3: Local decision of consecutive three deletions in a repetitive region. Upper: Bowtie2's local alignment decision. In many alignments, the two deletions on the left are not evident, and instead a 2 bp insertion is introduced to the left of the deletions. This creates a shift in reference coordinates that prevents biastools' context-aware assignment method from assigning the reads correctly. Lower: The different local alignment decisions made by BWA MEM and Bowtie2 using the same set of the reads.

Figure S4: In the IGV zoom out / zoom in view of the same region, the upper track is the BWA MEM default setup while the lower track is the BWA MEM with clipping penalty set to 30 instead of default 5. All the reads contain the 39 bp insertion are clipped in the default BWA MEM setup, thus create the biases toward the ALT allele. The BWA MEM with -L 30 behaves more like an end-to-end aligner, thus can handle the insertion without clipping.

Figure S5: An example of the low coverage result of LevioSAM 2 and direct-to-GRC methods. The top two panels show how the reads aligned by the two methods. The bottom three panels show the biased regions called by *biastools* scan in direct-to-GRC, LevioSAM 2, and the range of the centromeric region. While LevioSAM 2 did not classify the region on the left as biased, this is due simply to the low coverage in that region.

Table S1: Number of balanced sites and different categories of biased sites on chromosome 16. The simulated WGS reads of HG002 are aligned by 8 different tools. The best results of balanced and Bias “Loss” are marked in bold and underline. The second best results are marked in bold.

		Bowtie 2	BWA-MEM	BWA-MEM (-L 30)	Minimap2	Giraffe-linear	Giraffe-major	Giraffe-pop5	Giraffe-1KGP
SNV	Balanced	69847	69824	70073	69855	70032	70034	70106	70102
	Bias (Loss)	3369	3306	3060	3291	3129	3104	3008	2980
	Bias (Flux)	963	1047	1054	1035	981	984	989	1013
	Bias (Local)	139	167	166	164	185	211	229	234
	Outliers	138	112	103	111	129	123	124	127
Gap	Balanced	14061	13723	14229	13774	13952	13989	14007	14046
	Bias (Loss)	1151	1319	878	1291	1155	1113	1083	1052
	Bias (Flux)	212	287	278	274	252	249	245	247
	Bias (Local)	136	223	167	212	194	202	221	214
	Outliers	31	39	39	40	38	38	35	32